# Sensitivity of the polar boundary layer to transient phenomena

**Amandine Kaiser**[1], **Nikki Vercauteren**[1,2], **and Sebastian Krumscheid**[3]

[1]Department of Geosciences, University of Oslo, Oslo, Norway
[2]Institut für Geophysik und Meteorologie, Universität zu Köln, Cologne, Germany
[3]Department of Mathematics, Karlsruhe Institute of Technology, Karlsruhe, Germany

**Correspondence:** Amandine Kaiser (amandine.kaiser@geo.uio.no)

**Abstract.** Numerical weather prediction and climate models encounter challenges in accurately representing flow regimes in the stably stratified atmospheric boundary layer and the transitions between them, leading to an inadequate depiction of regime occupation statistics. As a consequence, existing models exhibit significant biases in near-surface temperatures at high latitudes. To explore inherent uncertainties in modeling regime transitions, the response of the near-surface temperature inversion to transient small-scale phenomena is analyzed based on a stochastic modeling approach. A sensitivity analysis is conducted by augmenting a conceptual model for near-surface temperature inversions with randomizations that account for different types of model uncertainty. The stochastic conceptual model serves as a tool to systematically investigate which types of unsteady flow features may trigger abrupt transitions in the mean boundary layer state. The findings show that the incorporation of enhanced mixing, a common practice in numerical weather prediction models, blurs the two regime characteristic of the stably stratified atmospheric boundary layer. Simulating intermittent turbulence is shown to provide a potential workaround for this issue. Including key uncertainty in models could lead to a better statistical representation of the regimes in long-term climate simulation. This would help to improve our understanding and the forecasting of climate change in high-latitude regions.

## 1 Introduction

The polar and nocturnal stably stratified boundary layer (SBL) is typically classified into two distinct regimes: a weak (wSBL) one and a very stable (vSBL) one. The wSBL is characterized by a well-defined boundary layer with relatively continuous turbulence in both space and time, and it typically occurs with either cloud cover or moderate to strong winds (Mahrt, 2014). On the other hand, the vSBL is defined by strong stratification and weak winds. A sharp transition is found between those two regimes, occurring in a narrow range of wind speeds. The existence of the two distinct regimes has been shown in many studies, ranging from observational studies (Mahrt, 1998; Vignon et al., 2017) to conceptual modeling (McNider, 1995; van de Wiel et al., 2017) and turbulence-resolving numerical simulations (Ansorge and Mellado, 2014; Donda et al., 2015).

Conceptual models have been used to understand regime transitions and to explore the relevant feedback processes which can explain the existence of the two regimes. Using a two-layer numerical model that represents the exchanges between the surface and the SBL, McNider (1995) demonstrated the presence of bistable equilibria within the system. Specifically, they showed that both wSBL and vSBL could be possible solutions for a given forcing of the SBL. Consequently, the system has the capability to undergo transitions between the two regimes due to the effects of random perturbations of the equilibrium state. As another example, the conceptual model suggested by van de Wiel et al. (2017) uses a surface energy balance model combined with a bulk parameterization of the turbulent heat transport and a simple parameterization of the soil and radiative transfer to explore the nonlinear feedback process related to the energy fluxes coupling

at the land–atmosphere interface. The system represented in the conceptual model can lead to a bistable system; i.e., in some cases, for the same wind forcing, both the vSBL and wSBL are possible solutions. The authors showed that the system is bistable for a range of different parameters. Moreover, they found that the scatter seen in the bistable region of observational data can be explained by the fact that perturbations away from the equilibrium are largely undamped. In our study, we investigate when perturbations of the dynamics or their forcing parameters lead to a nonlinear feedback that induces a regime transition. For this study, we will use model parameters which were estimated based on polar measurements as bistability is especially relevant in the polar context (van de Wiel et al., 2017). The existence of multi-valued solutions has strong implications for the predictability of the SBL since small perturbations or uncertainty in initial and forcing conditions may lead to very different outcomes of the mean state of the SBL. In particular, for NWPs, the forecast uncertainty is significantly impacted by the spread of possible outcomes related to the before-mentioned bistability. In addition, for Earth system models, it is of high relevance to have a correct statistical representation of the two regimes to accurately model seasonal heat fluxes between the surface and the atmosphere.

Several observational studies have shown the abrupt character of transitions between the two stable states (Baas et al., 2019; Vignon et al., 2017). These transitions can be controlled by changes in the large-scale forcing. Based on observational data, Acevedo et al. (2019) connected regime transitions from vSBL to wSBL with a change in wind direction first from the ocean and then from land. They postulated that the transition is related to a change in horizontal advection patterns. Moreover, Abraham and Monahan (2019) showed, by applying a hidden Markov model to cluster observational data from multiple locations in two regimes, that transitions from wSBL to vSBL most likely occur shortly after sunset, hence relating them to the onset of nocturnal radiative cooling. The statistical study highlighted no regularity for vSBL-to-wSBL transitions, which points to possibly small-scale random features at the origin of transitions. Indeed, transient phenomena can in some cases be sufficient to induce a large-scale change in the SBL. Our hypothesis is that transitions can be due to noise-induced tipping, for example following the influence of an isolated turbulent burst that can trigger a nonlinear feedback process where turbulence is eventually regenerated in the entire boundary layer. Direct numerical simulations have shown that, after a localized, random perturbation of the flow, transitions from vSBL to wSBL states can occur (Donda et al., 2015). Field studies have also revealed examples of transitions triggered by small-scale perturbations (Sun et al., 2012; Lan et al., 2022). The following numerical study analyzes the sensitivity of the SBL to small-scale variability in the forcing and in the turbulent fluxes.

To analyze which small-scale perturbations can nonlinearly amplify and lead to regime transitions, a sensitivity study is designed based on a stochastic extension of the conceptual model by van de Wiel et al. (2017). In a bistable system, abrupt transitions or tipping points can be noise induced, meaning that a transient perturbation can be enough to trigger a regime transition (Ashwin et al., 2012). These noise-induced transitions occur without passing a critical value of the control bifurcation parameter, here the ambient wind velocity. Different types of noise are added to the model to represent transient phenomena. With this randomized model, we study the sensitivity of regime transitions to perturbations of the temperature dynamics and wind speed. Kaiser et al. (2020) successfully tested a statistical indicator for the dynamical stability with a similarly randomized version of the model by van de Wiel et al. (2017). Moreover, Abraham et al. (2019) developed a stochastic model with stratification-dependent transition probabilities, and Ramsey and Monahan (2022) defined a data-driven stochastic differential equation model for temperature inversion variability. Here, the goal is not to derive a stochastic model but rather to analyze the sensitivity of the boundary layer dynamics to inform future model developments. The focus hereby is on the polar night, and therefore, the model parameter values are chosen such that they represent a polar context.

Among the small-scale phenomena that have been shown to be relevant for regime transitions, turbulence is described by a model closure. This model structure typically relies on Monin–Obhukov similarity theory (MOST), which assumes stationarity of the turbulent statistics and hence does not represent intermittency and turbulent bursts (Foken, 2006). This assumption is known not to hold in the vSBL (Liang et al., 2014; Mahrt and Bou-Zeid, 2020). In MOST, the closure makes use of a stability function that scales the mixing length scale, adjusting the amount of turbulent mixing to the dimensionless static stability of the flow (Cuxart et al., 2005). Numerical weather prediction models (NWPs) use different types of stability functions, and several works have shown that the choice impacts the representation of SBL regimes. Commonly, NWPs are formulated such that they include enhanced mixing to prevent excessive surface cooling (Sandu et al., 2013), also called runaway cooling. This is implemented through the use of a stability function which ensures that turbulence is artificially sustained even under very stable conditions. This type of stability function is called a long-tail stability function. Its use is often justified by the need to account for contributions to vertical mixing associated with surface heterogeneity, gravity waves, or mesoscale variability that are not explicitly represented in models (Sandu et al., 2013). However, these types of stability functions smooth transitions; i.e., they prevent abrupt transitions (Baas et al., 2017; van de Wiel et al., 2017). On the other hand, Kähnert (2022) showed that, when a short-tail stability function, which suppresses turbulence in very stable conditions,

is used, models can become locked in a vSBL state until sufficiently strong forcing induces a transition.

In their paper, van de Wiel et al. (2017) considered three types of stability functions, namely the cutoff, short-tail, and long-tail forms. In all cases, the bistability of the SBL was observed for some parameter values, but the structure of the bifurcation diagram varied significantly depending on the stability function. To include model uncertainty related to the stability function, wind forcing, or unresolved processes, the conceptual model by van de Wiel et al. (2017) is transformed from a deterministic model to a stochastic one. This stochastic model can then be reinterpreted as a stochastic gradient flow in which the temperature inversion evolves according to an underlying energy potential, and the equilibrium points correspond to minima of this potential. If several minima exist, then the different equilibria are separated by a local maximum that can be interpreted as a potential barrier that needs to be crossed if the system is to transition between the different equilibria. We highlight that the choice of the stability function has a large impact on the height of the potential barrier and hence on the energy input needed to undergo a regime transition. Therefore, the likelihood of noise-induced regime transitions in a model is highly sensitive to the choice of stability function. Although this is motivated in this study in a highly simplified, conceptual model, this may have consequences in NWPs as the variability of unresolved degrees of freedom can induce abrupt regime transitions. This effect is lost with a long-tail stability function where the potential barrier is shallow. We hypothesize that the combination of the variability of unresolved scales, through stochastic modeling, with a short-tail stability function may be necessary to enable sharp transitions. This is particularly relevant in polar regions where the long-lived polar night can lead to a flow that possibly evolves in the vicinity of the transition region for a long time. Enabling sharp transitions may be relevant for correctly estimating long-term statistics of surface fluxes. One way of including the variability of unresolved degrees of freedom is by exploring stochastic parameterizations (Berner et al., 2017). An example of a stochastic parameterization of turbulence is the stochastic stability function which is defined by Boyko and Vercauteren (2023b). The authors proposed a stochastic extension and implemented it in MOST for a single-column model that can represent unsteady and intermittent turbulence. Similarly, to qualitatively reproduce temporally localized bursts of turbulence in the vSBL regime, we use a stochastic parameterization of the short-tail stability function in the form used by van de Wiel et al. (2017). This model randomization is used to study to what extent localized events can trigger regime transitions in the stable boundary layer.

In summary, we want to study the sensitivity of the SBL to perturbations that represent model uncertainty. Model uncertainty arises, among other things, due to unresolved processes in the model and forcing uncertainty. In addition, surface heterogeneity and non-stationary turbulent variables contradict the assumptions of MOST, leading to uncertainty in the turbulence parameterization. We utilize a randomized version of the nonlinear model suggested by van de Wiel et al. (2017) with model parameters representing a polar context to address the following research questions:

- To what extent does the choice of stability function affect the likelihood of a transition?

- How sensitive are regime transitions to different types of model uncertainty?

- How does the inclusion of turbulent bursts in the vSBL regime affect the timing of regime transition?

To tackle these questions, in Sect. 2.1, the model by van de Wiel et al. (2017) is introduced in detail. Section 2.2.1 discusses the impact of the choice of stability function on the representation of regimes and regime transitions. Following that, Sect. 2.2.2 presents a sensitivity analysis of regime transitions, analyzing the impact of model uncertainty due to small-scale fluctuations of unresolved processes. Additional randomization is then investigated in Sect. 2.2.3, where the dynamical forcing variable, i.e., the geostrophic wind, is treated as a random variable. As the simplest scenario, we use an Ornstein–Uhlenbeck process to represent the random geostrophic wind forcing. Lastly, Sect. 2.2.4 studies the impact of a randomized stability function. For this, the short-tail stability function by van de Wiel et al. (2017) is extended with multiplicative noise. We study how the timing of transitions changes in relation to the wind forcing with a randomized stability function.

## 2 Conceptual model for near-surface temperature inversions

### 2.1 Model description

In their study, van de Wiel et al. (2017) developed a conceptual model that describes the evolution of near-surface inversion strength over time. To achieve this, they combined a surface energy budget with a bulk model of the lower atmosphere and investigated near-surface temperature inversion transitions in the polar and nocturnal atmospheric boundary layer. The model highlights a nonlinear relationship between the stability of temperature inversion and ambient wind speed by employing a first-order ordinary differential equation to represent the time evolution of the temperature differences, $\Delta T$, between a reference height ($T_r$) and the surface temperature ($T_s$). This temperature difference is referred to as temperature inversion in the rest of the paper. Based on the observation that the wind speed was constant at a certain height (Hooijdonk et al., 2015), the model assumes that the wind speed and temperature are constant at this reference height, $z_r$. A brief summary of the model is given below, with more details available in van de Wiel et al. (2017). Assuming a

symmetry between the bulk temperature differences in the atmosphere and soil, the evolution of the temperature inversion is governed by a surface energy balance:

$$\frac{\mathrm{d}\Delta T}{\mathrm{d}t} = \frac{1}{c_\mathrm{v}}(Q_\mathrm{n} - G - H), \tag{1}$$

where $c_\mathrm{v}$ TS1 is the heat capacity of the surface, $Q_\mathrm{n}$ is the net longwave radiative flux, $G$ is the soil heat flux, and $H$ is the turbulent sensible heat flux. After parameterizing the fluxes, the model takes the following form:

$$\frac{\mathrm{d}\Delta T}{\mathrm{d}t} = \frac{1}{c_\mathrm{v}}(Q_\mathrm{i} - \lambda\Delta T - \rho c_\mathrm{p}c_\mathrm{D}U\Delta T f_\mathrm{stab}(R_\mathrm{b})),$$

$$R_\mathrm{b} = z_\mathrm{r}\frac{g\Delta T}{T_\mathrm{r}U^2}, \tag{2}$$

where $R_\mathrm{b}$ is the bulk Richardson number, $Q_\mathrm{i}$ represents the isothermal net radiation, $\lambda$ is the net linear effect of all feedbacks from soil heat conduction and radiative cooling represented with a lumped parameter, $\rho$ stands for the density of air at constant pressure, and $c_\mathrm{p}$ is the heat capacity of air at constant pressure. Additionally, $c_\mathrm{D} = (\frac{\kappa}{\ln(z_\mathrm{r}/z_0)})^2$ is referred to as the neutral drag coefficient where $\kappa$ is approximately equal to 0.4 and the von Kármán constant, $z_0$ is the roughness length, and $z_\mathrm{r}$ is the reference height. Moreover, $U$ represents the wind speed at height $z_\mathrm{r}$. The sensible heat flux $H$ is parameterized with the Monin–Obukhov similarity theory, which uses a stability function, $f_\mathrm{stab}(R_\mathrm{b})$, to describe how much turbulence is present in relation to the strength of the stratification. The same values as in van de Wiel et al. (2017) (Table 1, Dome C) are used for all parameters unless stated otherwise. The focus is on parameters representing a polar context as studies have shown that, in these regions, bistability exists (Ramsey and Monahan, 2022; van de Wiel et al., 2017), and hence, transitions can be more abrupt. Therefore, in this context, the impact of small-scale perturbations on regime transitions is especially relevant. For convenience, all parameter values are summarized in Table 1. The Dome C data were collected at the Concordia Research Station located on the Antarctica Plateau. This French–Italian research facility, situated at an elevation of 3233 m a.s.l., is described in detail by Genthon et al. (2010). The data from 2017, consisting of 10 min averaged meteorological data, are, for example, analyzed by Vignon et al. (2017) and Baas et al. (2019). In our study, the focus lies on the polar night, which lasts from March to September, and the following parameters are crucial: the temperature at 9.4 m height and the surface, the wind speed (m s$^{-1}$) at 8 m height, and the radiation in the polar night. Following van de Wiel et al. (2017), only the data where the radiative forcing ($R^+ = \mathrm{SW}^\downarrow - \mathrm{SW}^\uparrow + \mathrm{LW}^\downarrow$) is less than 80 W m$^{-2}$ are considered when the data are studied. $\mathrm{SW}^\downarrow$ is the incoming shortwave radiation, $\mathrm{SW}^\uparrow$ is the outgoing shortwave radiation, and $\mathrm{LW}^\downarrow$ is the incoming longwave radiation. For $R^+ < 80$ W m$^{-2}$, the sky is very clear (Vignon et al., 2017), and radiative cooling is pronounced.

**Table 1.** Default parameter values for the model (i.e., Eq. 2).

| Parameter | Value | Unit |
|---|---|---|
| $Q_\mathrm{i}$ | 50.0 | W m$^{-2}$ |
| $\lambda$ | 2.0 | W m$^{-2}$ K$^{-1}$ |
| $\kappa$ | 0.4 | – |
| $c_\mathrm{v}$ | 1000 | J m$^{-2}$ K$^{-1}$ |
| $\rho$ | 1.0 | kg m$^{-3}$ |
| $c_\mathrm{p}$ | 1005.0 | J kg$^{-1}$ K$^{-1}$ |
| $z_0$ | 0.01 | m |
| $z_\mathrm{r}$ | 10.0 | m |
| $U$ | 5.2 | m s$^{-1}$ |
| $g$ | 9.81 | m s$^{-2}$ |
| $T_\mathrm{r}$ | 243.0 | K |
| $\alpha$ | 5.0 | – |

The designers of the model (Eq. 2) provide an in-depth analysis of the model's equilibrium states and their stability against perturbations. For the purpose of this study, the most important features of the model are that the solution is bistable with the above-mentioned parameters. This means that, for specific wind speeds ($U$), the model has two stable equilibrium solutions and an unstable one. Transitions between stable states can be triggered by large-enough perturbations that force the system to cross the potential barrier. Figure 1c shows the locations of the stable states for the discussed model driven by the parameters given in Table 1. The blue line is a plot of the equilibrium solutions of the parameterized model by van de Wiel et al. (2017) (i.e., Eq. 2). For low wind speeds and high temperature differences, there is one single equilibrium. The same is true for high $U$ and small $\Delta T$. The two corresponding equilibrium branches are marked with a solid line. In between those two stable regimes, there is a range of $U$ values highlighted in red with two stable equilibria (solid lines) separated by an unstable equilibrium (dotted line). Connecting this to the SBL context, the first stable regime is one with very stable stratification, while the other one is weakly stable. As noted by van de Wiel et al. (2017), a similar behavior is apparent in observational data, for example, measured at Dome C. When plotting the difference between temperature measured at 9.4 m and the surface, i.e., $\Delta T = T_{9.4\,\mathrm{m}} - T_\mathrm{s}$, against the wind speed measured at 8 m, $U_{8\,\mathrm{m}}$, a back-folding of the points becomes discernible when $R^+ < 80$ W m$^{-2}$ (as shown in Fig. 6 in van de Wiel et al. (2017) and less clearly in our Fig. 1c). In Fig. 1c, the orange dots are the described 10 min averaged observational data from Dome C. In the data, a weakly stable regime is clearly observable, but the very stable regime is not as distinct. As shown by van de Wiel et al. (2017) and in Fig. 1c, the model provides a qualitative representation of the data at Dome C, particularly regarding the existence of two limiting states. Based on that, the model is chosen to study transitions in the polar SBL.

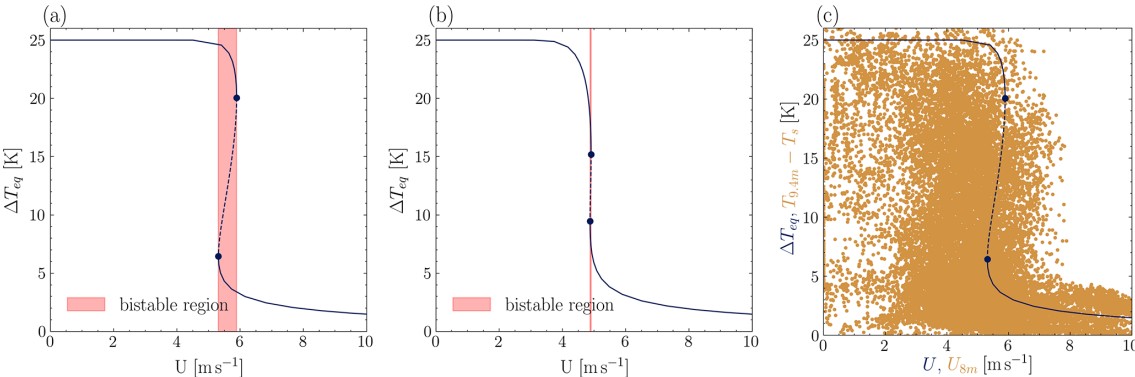

**Figure 1.** Equilibrium points of the model with a **(a)** short-tail and **(b)** long-tail stability function plotted over wind speed. The dotted lines mark unstable equilibria, while the solid lines correspond to stable ones. The red region is the region with two possible solutions for the same forcing conditions. In plot **(c)**, the orange dots are observational data from Dome C, and the blue line corresponds to the equilibrium solutions of the model with a short-tail stability function.

## 2.2 Randomization strategies

Three strategies are employed to address model uncertainty in the conceptual model proposed by van de Wiel et al. (2017) (Eq. 2). Firstly, to account for uncertainty arising from unresolved processes within the model, a stochastic differential equation (SDE) version of the model is presented in Sect. 2.2.2. Moreover, uncertainty related to the wind forcing is addressed in Sect. 2.2.3 by modeling the wind with an Ornstein–Uhlenbeck process. This section also investigates the combined effect of randomizing the wind forcing and the model itself. Lastly, the study explores a randomized stability function as a potential solution to account for uncertainty in the parameterization of turbulence. A stochastic stability function allows the inclusion of turbulent bursts even in very stable stratification. In all cases, the aim is to account for small-scale fluctuations rather than large-scale changes. As van de Wiel et al. (2017) showed that their model, in its deterministic form, is highly sensitive to a change in the lumped parameter $\lambda$, which represents soil and radiation feedbacks, and the surface roughness, $z_0$, both parameters are not further investigated in this study. Stochastic parameterizations have the merit to provide uncertainty estimations, but beyond that, they have the potential to induce regime transitions if the physical system has multiple coexisting equilibria. In that sense, they may be necessary for better representing the mean state of the system (Berner et al., 2017). Here, the stochastic modeling will be used to study the effect of small-scale fluctuations of an unresolved process in the model and of the forcing wind speed or to include localized turbulent bursts. The small-scale fluctuations of an unresolved process are, for example, included in the model by the addition of noise. This new randomized model is defined as

$$d\Delta T = \frac{1}{c_v}(Q_i - \lambda \Delta T - \rho c_p c_D U \Delta T f_{stab}(R_b))dt$$
$$+ \sigma_i \, dW_i \, ,$$

$$\Delta T_0 = x_0, \tag{3}$$

where $W_i$ is a Wiener process. In Sect. 2.2.2, a sensitivity analysis will be carried out to comprehend and discern the impacts of the aforementioned model uncertainties. Through the utilization of the sensitivity analysis, we aim to enhance our comprehension of which of these model uncertainties have a large impact on the statistical representation of regimes and transitions between them. Due to the presence of noise, the equilibrium points of the deterministic and random models will not be exactly the same, but the dominant effects will be. The simulation time for every model run should be long enough to reach a quasi-equilibrium state. To achieve this, rather than simulating until a temporal statistical equilibrium is reached, the Monte Carlo sampling study is performed. As the idealized model is forced by a constant wind speed which would vary with synoptic conditions and not be constant for more than a few hours, we deemed a simulation time of 24 h with time steps of 1 s to be a decent compromise. In addition, this choice was also for practical reasons as the focus in the following sections is on a grid search combined with Monte Carlo simulations, which requires a significant amount of computing power. The Monte Carlo simulations are run with 500 realizations. A comparison with 1000 simulations showed similar results. Hence, 500 realizations are deemed to be sufficient for the rest of the sensitivity analysis. The model parameters are the same as in Table 1. All SDEs are solved using the function itoint from the Python library sdeint by Aburn (2017). This function applies the stochastic Runge–Kutta algorithm of order 1.0 (Rößler, 2010).

### 2.2.1 Impact of the choice of the stability function

In MOST, the strength of turbulence is scaled by the mixing length, which is itself adjusted through a stability function (Foken, 2006). This stability function corrects the turbulence strength depending on the dimensionless stability of

the flow. The choice of stability function has a large impact on the parameterization of turbulence. Under strong stratification, long-tail stability functions allow turbulence to exist, while with a short-tail stability function, turbulence is largely suppressed after a critical Richardson number. Long-tail stability functions are typically utilized in NWPs to avoid excessive cooling in highly stable conditions within the SBL (Sandu et al., 2013; van de Wiel et al., 2002). The use of such stability functions, however, has broader consequences and strongly influences the representation of transitions between SBL regimes. By artificially sustaining turbulence, even under very stable conditions, the transitions become less abrupt (Baas et al., 2017). Further impacts of the stability function on the representation of the bistability of the system are analyzed here, focusing on the likelihood of regime transitions when model uncertainty is present. Analogously to van de Wiel et al. (2017), we use $f_{stab}(R_b) = \exp(-2\alpha R_b)$ as a long-tail stability function and $f_{stab}(R_b) = \exp(-2\alpha R_b - (\alpha R_b)^2)$ as a short-tail one, with $\alpha = 5$. Both short-tail and long-tail stability functions are plotted in Fig. 2. For a Richardson number larger than 0.35, the short-tail stability function approaches zero. Not all degrees of freedom are represented in the conceptual model by van de Wiel et al. (2017), leading to inherent model uncertainty. To account for this, the model is transformed from a deterministic one to a stochastic one using three different types of randomization, which are described in detail in Sect. 2.2.2, 2.2.3, and 2.2.4. The likelihood of transitioning between the two stable states, in the context of noise-induced tipping, is very dependent on the choice of stability function. There are two main reasons for that. Firstly, the bistable range of wind speeds is much narrower when using the long-tail stability function. This can be seen in Fig. 1. The bistable range is the region in which the model can have two stable solutions for the same forcing, i.e., $U$. For the short-tail stability function, the bistable region spans from $U = 5.31$ to $5.89\,\mathrm{m\,s^{-1}}$ (Fig. 1a), while for the long-tail stability function, it is between 4.87 and $4.9\,\mathrm{m\,s^{-1}}$ (Fig. 1b). That means that the bistable region for the long-tail stability function has only 6 % of the width of the one for the short-tail stability function. Therefore, with a long-tail stability function, it is significantly less likely to transition from one stable regime to the other. This agrees with the findings of Baas et al. (2017) that enhanced mixing, which is the effect of a long-tail stability function, is detrimental for modeling the SBL. For example, near-surface cooling and wind shear are systematically underestimated with the enhanced mixing model. Secondly, to explain why the likelihood of transition is very dependent on the choice of stability function, we introduce the concept of a potential. For this, Eq. (2) can be rewritten as a gradient system:

$$\frac{\mathrm{d}\Delta T}{\mathrm{d}t} = -V'(\Delta T), \quad \Delta T(t_0) = \Delta T_0,$$

where $V : \mathbb{R} \to \mathbb{R}$ is an underlying energy potential influencing the dynamics of the temperature inversion. The extrema of the potential $V$ correspond to the equilibria of $\Delta T$; i.e., for an equilibrium point $\Delta T_e$, it holds $V'(\Delta T_e) = 0$. In general, the dynamics of $\Delta T$ will evolve towards the nearest local minimum of the potential. If it resides there, signifying a stable equilibrium, it would require the addition of significant random fluctuations to exit this state. Indeed, if a local maximum separates two local minima, i.e., two possible stable equilibria, the difference between the potential's minimum and maximum is an energy depth that the dynamics have to overcome in order to transition to a second stable equilibrium. This is called a potential barrier. In our context, that means that, if the wind forcing is such that both vSBL and wSBL are supported solutions, the potential barrier describes the intensity of the fluctuations of $\Delta T$ that are needed to transition between the two states, assuming no other changes in the forcing or dynamics. Using Fig. 3, we can compare the potentials for a short-tail stability function (Fig. 3a) and a long-tail one (Fig. 3b). The lines correspond to potentials for different wind speeds which are in the bistable region. The green histograms are the results of 1000 simulations of the model, where the model itself is randomized (Eq. 3). For a detailed description of the randomization, see Sect. 2.2.2. All simulations were started in the very stable regime, i.e., $\Delta T_0 = 24\,\mathrm{K}$, and the simulation time was 24 h for all simulations. In the short-tail stability function histogram, the two stable equilibria ($\Delta T = 4$ and 24 K) distinctively show up, and there are clearly multiple transitions between both regimes. In contrast, for the long-tail stability function, the very stable regime is not distinctively separated from the unstable one (located at $\Delta T = 12\,\mathrm{K}$). This is related to the fact that the potential barrier is much shallower with the long-tail stability function, which can be seen by comparing the histograms of the two stability functions (Fig. 3a and b). Even though there are multiple transitions between the two stable regimes with the long-tail stability function, the system rarely stays in the very stable regime. Based on these results, we hypothesize that, by using a short-tail stability function augmented by random fluctuations with locally enhanced mixing instead of the averaged enhanced mixing of the long-tail stability function, transitions are better represented. In Sect. 2.2.4, the stochastic stability function is introduced. This randomized parameterization accommodates the representation of transient bursts of turbulent mixing, which could force the system to transition.

### 2.2.2 Model sensitivity to internal variability

Many processes are not resolved in the simplified conceptual model by van de Wiel et al. (2017) (Eq. 2). In a first randomization strategy, a stochastic term is added to the model to represent small-scale fluctuations of the dynamics of the temperature inversion due to unresolved processes. The goal is to quantify the impact of model uncertainties as an additive-noise component on the statistical representation of regimes in the SBL. The model has been defined in Sect. 2.2, but as a

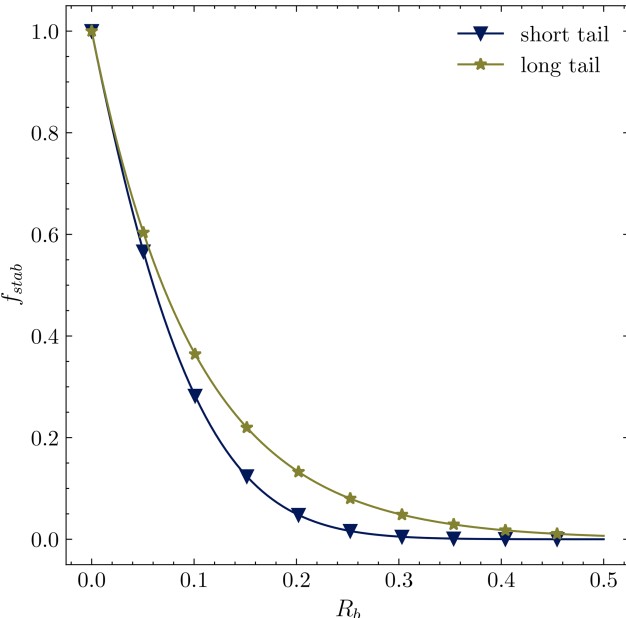

**Figure 2.** Long-tail and short-tail stability functions considered for the model plotted over the Richardson number.

recap, the equations are repeated here:

$$d\Delta T = \frac{1}{c_v}(Q_i - \lambda \Delta T - \rho c_p c_D U \Delta T f_{\text{stab}}(R_b))dt$$
$$+ \sigma_i dW_i ,$$
$$\Delta T_0 = x_0, \qquad (4)$$

5 where $W_i$ is a Wiener process (i.e., stochastic process), $\sigma_i$ scales the fluctuation intensity, and $x_0$ is either equal to 4 K (wSBL) or 24 K (vSBL).

To assess the model's response to this randomization, a sensitivity analysis is conducted, and the focus is placed on 10 the representation of regime transitions (see Fig. 4). The adapted Eq. (4) of the model Eq. (2) is run 500 times for each combination of a range of different $U$ and $\sigma_i$ values. To distinguish the effect of the randomization on the two transition types, the simulations are started in the vSBL state, i.e., 15 $\Delta T_0 = 24$ K, and in the wSBL one, $\Delta T_0 = 4$ K. Then, for every $U$, the minimal $\sigma_i$, for which at least 80 % of all 24 h simulations include at least one transition, is identified. For convenience, this minimal $\sigma_i$ is abbreviated as $\sigma_{i,\text{min}}$. As anticipated, the value of $\sigma_{i,\text{min}}$ required to achieve transitions from 20 wSBL to vSBL is lower for low wind speeds and higher for high winds. Conversely, for transitions from vSBL to wSBL, the opposite trend is observed. This phenomenon can be explained by examining the plot of the equilibrium points for the deterministic model (Eq. 2) (see the dotted red line in 25 Fig. 4). At low wind speeds, the system has a single equilibrium state, which is the vSBL. Consequently, no noise is necessary to transition from wSBL to vSBL, while a higher noise amplitude is required to exit vSBL and transition to

wSBL. In the bistable region, two stable equilibrium states are present for the same $U$ value. In the first segment of the 30 bistable region, the wSBL and vSBL $\sigma_{i,\text{min}}$ converge until they are nearly identical for $U = 5.6$ m s$^{-1}$. At this point, the unstable equilibrium state is positioned approximately midway between the two stable states. Consequently, the same noise magnitude is required for transitions in both directions. 35 Subsequently, as the wind speed increases, the two $\sigma_{i,\text{min}}$ diverge again. Here, the reverse argument can be made in comparison to low wind speeds: no noise is necessary to transition to the wSBL state since it represents the sole equilibrium state of the system, and higher noise values are required to 40 exit this state and transition to the vSBL state. As the bifurcation diagram is not exactly symmetric, higher noise levels are required for wSBL-to-vSBL transitions and high $U$ values than for vSBL-to-wSBL transitions and low $U$ values. This rationale is justified as small-scale turbulent bursts may 45 introduce enough mixing to force a transition to the wSBL state, whereas the inverse scenario does not hold true. As expected, no noise is required for the system to transition to its equilibrium state (vSBL for low wind speeds and wSBL for high wind speeds). However, to depart from the equilib- 50 rium state, higher noise amplitudes are necessary. Within the bistable region, the introduction of noise with small amplitudes allows transitions in both directions.

To illustrate the effect of the randomization in the model, an example is shown for the symmetry wind speed $U =$ 55 5.6 m s$^{-1}$. In this case, $\sigma_{i,\text{min}}$ is equal to 0.18 Ks$^{-1/2}$ for transitions from vSBL to wSBL, which is slightly larger than for transitions in the other direction. The simulation is started in the vSBL regime. Figure 5 displays an instance of one model run with $\sigma_i = 0.18$ K s$^{-1/2}$ and $U = 5.6$ m s$^{-1}$. The 60 stable equilibria of the deterministic model are indicated by two solid red lines, while the location of the unstable equilibrium is marked by a dotted line. This specific model run exhibits two transitions between the vSBL and wSBL regimes. It is important to note that, in the absence of noise, the sys- 65 tem would remain in the very stable regime and not undergo any transitions.

Lastly, to give an example of how the randomization affects the model result statistically, the histogram of $\Delta T$ for 500 model runs for $U = 5.6$ m s$^{-1}$ is shown in Fig. 6. For 70 plot (a), all the simulations are started in the vSBL, and for plot (b), all the simulations are started in the wSBL. The $\sigma_i$ are the corresponding $\sigma_{i,\text{min}}$ from Fig. 4. For both simulation types, the histogram shows a higher probability of being in the regime where the simulation started. This effect is es- 75 pecially pronounced for the simulations which started in the wSBL state.

### 2.2.3 Model sensitivity to fluctuating wind speed

A narrow wind speed range exists in which a sudden change in the temperature inversion can be observed (e.g., Baas et al., 80 2019; see also Fig. 1c). To investigate when small varia-

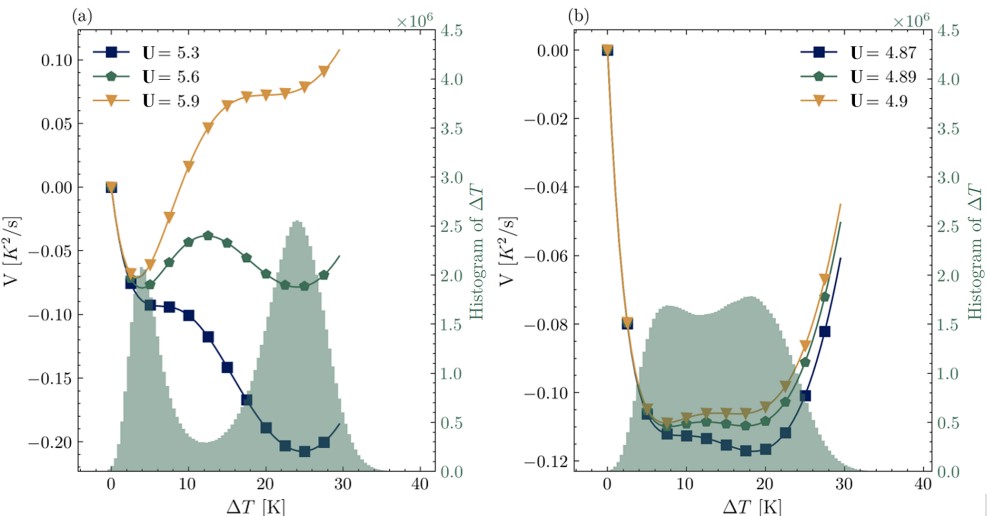

**Figure 3.** Potentials with a **(a)** short-tail and **(b)** long-tail stability functions for several wind speeds ($U$) and histograms of the results of 1000 simulations of the model with additive noise. In plot **(a)**, the mean of the random wind speed used to produce the histograms is equal to $5.6\,\mathrm{m\,s^{-1}}$, and in plot **(b)**, it is equal to $4.89\,\mathrm{m\,s^{-1}}$.

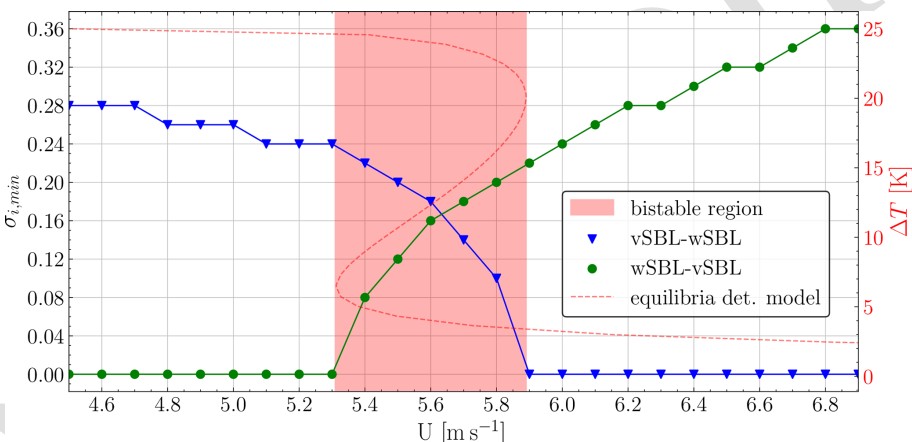

**Figure 4.** Results of the sensitivity study of the model (Eq. 4). For every $U$, the minimal $\sigma_i$ ($\sigma_{i,\min}$), for which at least $80\,\%$ of the 500 simulations have at least one transition of the indicated type, is marked. Simulations with an initial condition $\Delta T_0 = 24\,\mathrm{K}$ are plotted in green, and simulations with an initial condition $\Delta T_0 = 4\,\mathrm{K}$ are plotted in blue. The red line is the bifurcation diagram of the deterministic model (Eq. 2), and its bifurcation region is the red-shaded area.

tions of forcing wind speed can lead to sharp transitions, the conceptual model is modified by randomizing $U$ such that it fluctuates around a wind speed for which the system is bistable. To simulate the fluctuating wind speed, an Ornstein–Uhlenbeck process is incorporated into the model. This process is a widely used stochastic process in various applications (Pavliotis, 2014). The randomized model is defined as follows:

$$\mathrm{d}\Delta T = \frac{1}{c_\mathrm{v}}(Q_\mathrm{i} - \lambda\Delta T - \rho c_\mathrm{p} c_\mathrm{D} U \Delta T f_\mathrm{stab}(R_\mathrm{b}))\mathrm{d}t\,,\ \Delta T_0 = x_0\,,$$

$$\mathrm{d}U = -r(U - \overline{U})\mathrm{d}t + \sigma_U \mathrm{d}W_U\,,\quad U_0 = 5.6\,, \tag{5}$$

where $W_U$ is a Wiener process, and $r$ is a relaxation or mean reversion term. The value of $r$ is chosen to be $0.005\,\mathrm{s^{-1}}$ for

all simulations to achieve a mean reversion time of $200\,\mathrm{s}$, which is roughly the order of a submesoscale motion (Vercauteren et al., 2016). The value of $x_0$ is equal to either $4\,\mathrm{K}$ (wSBL) or $24\,\mathrm{K}$ (vSBL). The asymptotic mean of the Ornstein–Uhlenbeck process, $\overline{U}$, is set to $5.6\,\mathrm{m\,s^{-1}}$ as this is the middle of the bistable region (see Sect. 2.2.1). Different parameter values may give a different quantitative result, but we expect them to be qualitatively the same. The asymptotic variance of the Ornstein–Uhlenbeck process is $\mathbb{V}(U) = \frac{\sigma_U^2}{2r}$ (Pavliotis, 2014). The value for $\sigma_U$ is chosen based on $30\,\mathrm{min}$ averaged observational data from Dome C (Genthon et al., 2021). In their study, Baas et al. (2019) defined the bistable region for Dome C as $4\,\mathrm{m\,s^{-1}} \le U \le 7\,\mathrm{m\,s^{-1}}$. The same

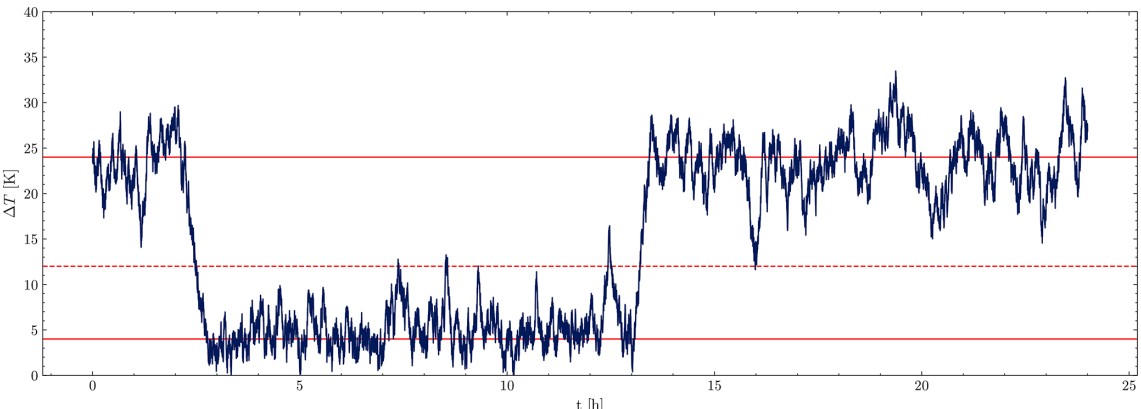

**Figure 5.** Plot of one solution of Eq. (4) with $U = 5.6\,\mathrm{m\,s^{-1}}$ and $\sigma_i = 0.18\,\mathrm{K\,s^{-1/2}}$. The dotted red line marks the unstable equilibria of the deterministic model, while the solid lines correspond to stable ones.

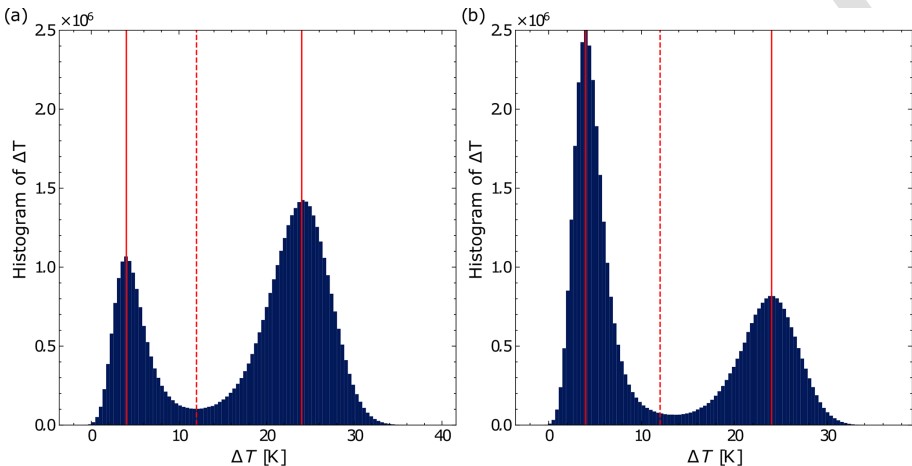

**Figure 6.** Histogram of all solutions of Eq. (4) with $U = 5.6\,\mathrm{m\,s^{-1}}$ and **(a)** $\sigma_i = 0.18\,\mathrm{K\,s^{-1/2}}$, $\Delta T_0 = 24\,\mathrm{K}$ and **(b)** $\sigma_i = 0.16\,\mathrm{K\,s^{-1/2}}$, $\Delta T_0 = 4\,\mathrm{K}$. The horizontal dotted red line marks the unstable equilibria of the deterministic model, while the solid lines correspond to stable ones.

thresholds and data from the year 2013, which has a mean value for the wind speed of $5.6\,\mathrm{m\,s^{-1}}$, are used to get an estimate of $\sigma_U$. To exclude mesoscale and longer timescales, the data are filtered with moving-average filtering, with a window length of 60 min. For this dataset, $\sigma_U$ is equal to $0.03\,\mathrm{m\,s^{-3/2}}$. Choosing $\sigma_U$ this big leads to high variations in $U$ and not to the small-scale perturbations we are interested in. In fact, while 97 % of all simulations of a Monte Carlo run with 500 simulations include a transition, 34 % of the wind velocity values are, on average, outside the bifurcation region, which contradicts the assumption of small-scale perturbations. One example of a run with $\sigma_U = 0.03\,\mathrm{m\,s^{-3/2}}$ is shown in Fig. 7b. Additionally, $\sigma_U = 0.01\,\mathrm{m\,s^{-3/2}}$ is considered as this is the smallest value for which hardly any of the wind velocity values are outside of the bifurcation region, i.e., on average, less than 1 %. But for this $\sigma_U$, none of the 500 simulations include a transition (see Fig. 7a).

As a next step, a model with both additive noise for internal variability and an Ornstein–Uhlenbeck process for wind velocity is considered. The Ornstein–Uhlenbeck process includes multiplicative noise. The model is defined as follows:

$$\mathrm{d}\Delta T = \frac{1}{c_v}(Q_i - \lambda \Delta T - \rho c_p c_D U \Delta T f_{\mathrm{stab}}(R_b))\mathrm{d}t + \sigma_i \,\mathrm{d}W_i\,, \quad \Delta T_0 = x\,,$$
$$\mathrm{d}U = -r(U - \overline{U})\mathrm{d}t + \sigma_U \,\mathrm{d}W_U\,, \quad U_0 = 5.6\,. \tag{6}$$

The relaxation parameter $r$ is the same as before. To quantify the impact of this model randomization, in terms of regime transitions, a sensitivity analysis is performed (see Fig. 8). The model (Eq. 6) is run 500 times for a combination of $\sigma_i$, $\sigma_U$, and $U$ values. For $\sigma_U$, the values 0, 0.01, and $0.03\,\mathrm{m\,s^{-3/2}}$ are chosen to allow a comparison of the results with the ones of the model where only the wind velocity is randomized (see Eq. 5). It will be noted that, when $\sigma_U = 0\,\mathrm{m\,s^{-3/2}}$, the model (Eq. 6) is behaviorally equal to the model (Eq. 4). The simulations are started in the vSBL

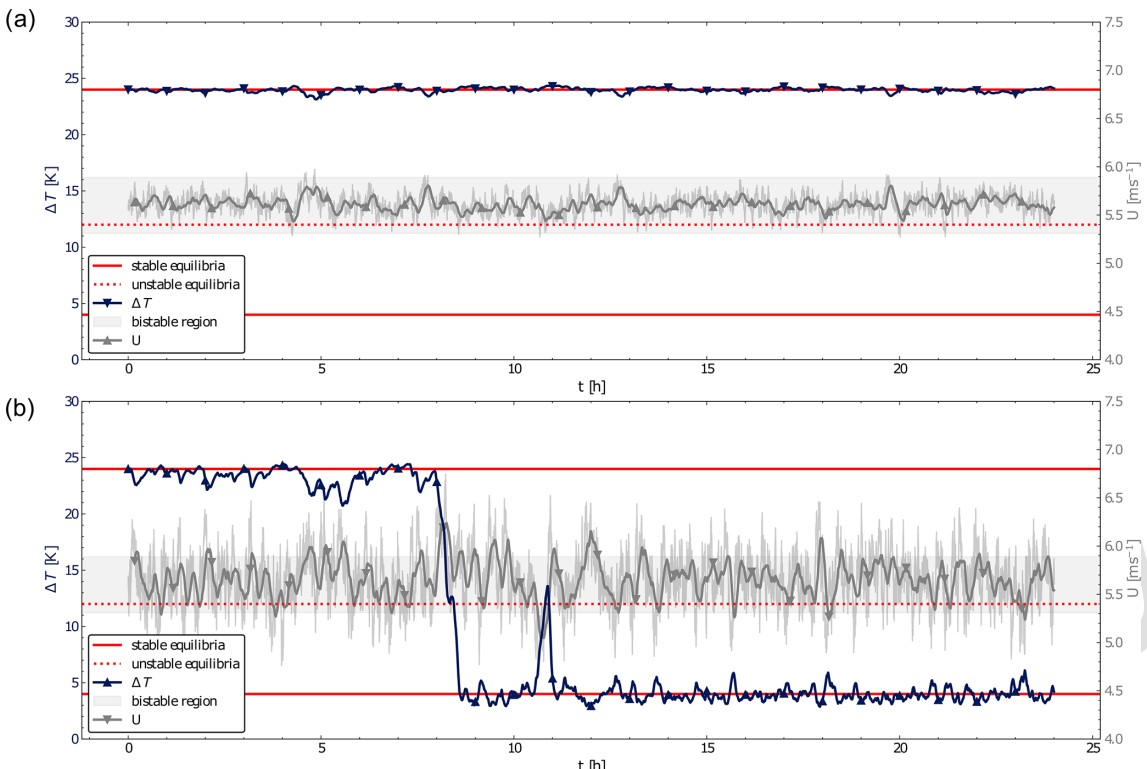

**Figure 7.** Plot of one solution of Eq. (5) (blue) and the corresponding wind speed $U$ (green), including its 10 min average (thick green line) for $\sigma_U = 0.01\,\mathrm{m\,s^{-3/2}}$ **(a)** and $\sigma_U = 0.03\,\mathrm{m\,s^{-3/2}}$ **(b)**. The dotted red line marks the unstable equilibria of the deterministic model, while the solid lines correspond to the stable ones. The green-shaded area is the bistable region for $U$.

state, $\Delta T_0 = 24\,\mathrm{K}$ (blue lines), and in the wSBL state, $\Delta T_0 = 4\,\mathrm{K}$ (green lines), to distinguish the effects on the two transition types separately. Then the minimal $\sigma_i$, for which at least 80 % of all simulations with the given $\sigma_U$ and $U$ values include at least one transition, is identified. This $\sigma_i$ is abbreviated as $\sigma_{i,\mathrm{min}}$.

It will be noted that the model with multiplicative noise may have different equilibrium points than the deterministic model (see Chap. 5.4 Pavliotis, 2014). This, for example, has been seen in Bashkirtseva et al. (2015) and Monahan (2002). Nonetheless, comparably with the results of the model (Eq. 4), where solely unresolved processes were considered while disregarding fluctuating wind speed, Fig. 8 shows that no noise is required for the system to transition to the equilibrium state of the deterministic model (vSBL for low wind speeds and wSBL for high wind speeds). However, to depart from the equilibrium state, higher noise amplitudes are necessary. In the bistable region, the introduction of noise with small amplitudes enables transitions in both directions, irrespective of the three different $\sigma_U$ values. Notably, larger $\sigma_U$ values permit smaller $\sigma_i$ values to induce transitions. The values of $\sigma_{i,\mathrm{min}}$ are identical or nearly identical for $\sigma_U = 0$ and 0.01.

Additionally, to provide a statistical representation of the model's randomization effects on the results, histograms

of $\Delta T$ for 500 model runs with $U = 5.6\,\mathrm{m\,s^{-1}}$ and $\sigma_U = 0.01\,\mathrm{m\,s^{-3/2}}$ are presented in Fig. 9. For plot (a), all the simulations are started in the vSBL, and for plot (b), all the simulations are started in the wSBL. The $\sigma_i$ are the corresponding $\sigma_{i,\mathrm{min}}$ from Fig. 8. The histograms reveal a higher probability of remaining within the initial regime for both simulation types, with a more pronounced effect observed for simulations starting in the wSBL state.

### 2.2.4 Model sensitivity to randomized stability function

In the conceptual model by van de Wiel et al. (2017) (Eq. 1), the sensible heat flux is parameterized using MOST, which uses the assumption that turbulence is stationary. In the vSBL regime, this assumption does not hold, and turbulence is rather intermittent and unsteady (Liang et al., 2014; Mahrt and Bou-Zeid, 2020). Therefore, in this section, the impact of uncertainties on the turbulence parameterization is studied. This is particularly significant as localized turbulent events can trigger regime transitions in the stable boundary layer (Lan et al., 2022). To qualitatively reproduce continuous bursts of turbulence, a model is introduced which enables temporally localized enhancement of turbulence. This is achieved by enhancing the mixing length. In MOST, the mixing length is adjusted through a stability correction func-

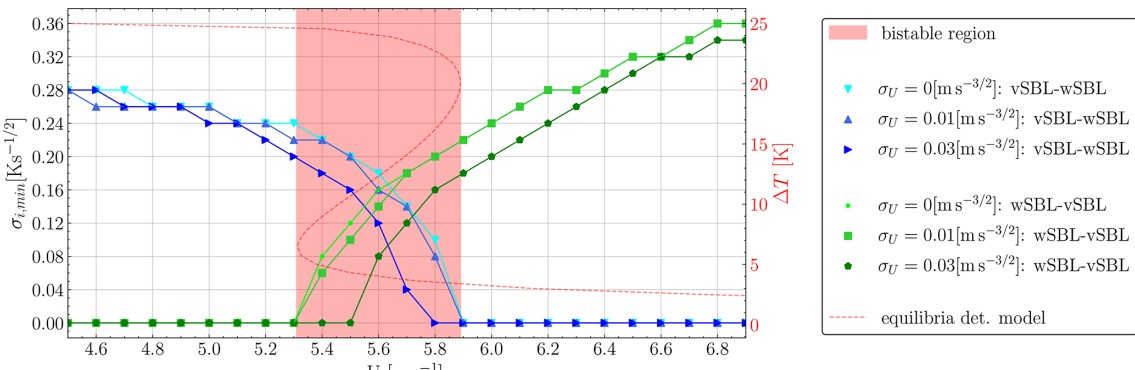

**Figure 8.** Results of the sensitivity study of the model (Eq. 6). For every $U$ the minimal $\sigma_i$ ($\sigma_{i,\min}$), for which at least 80 % of the 500 simulations have at least one transition of the indicated type, is marked. Simulations with an initial condition $\Delta T_0 = 24$ K are plotted in green, and simulations with an initial condition $\Delta T_0 = 4$ K are plotted in blue. The dotted red line is the bifurcation diagram of the deterministic model (Eq. 2), and its bifurcation region is the red-shaded area.

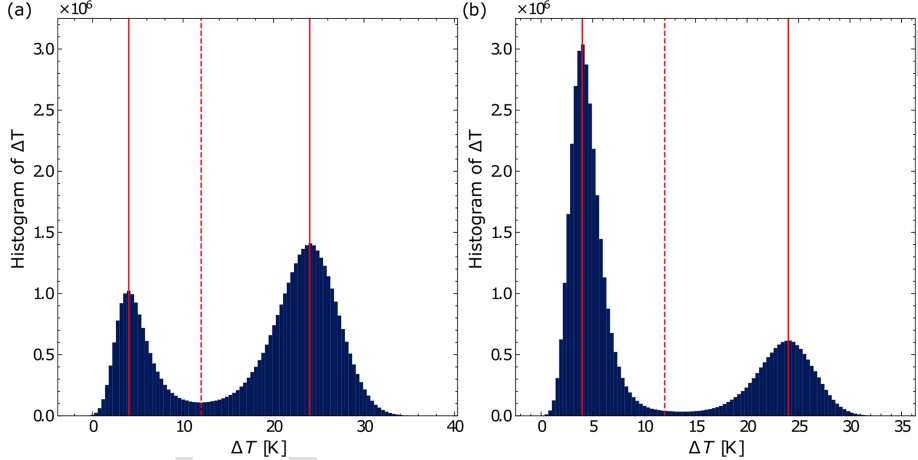

**Figure 9.** Histogram of all solutions of Eq. (6) with $U = 5.6 \, \mathrm{m \, s^{-1}}$, $\sigma_U = 0.01$ and **(a)** $\sigma_i = 0.18$, $\Delta T_0 = 24$ K and **(b)** $\sigma_i = 0.14$, $\Delta T_0 = 4$ K. The horizontal dotted red line marks the unstable equilibria of the deterministic model, while the solid lines correspond to stable ones.

tion. Therefore, a model is suggested where localized turbulence bursts are represented with a stochastic stability function. As the stability functions are the data-driven component of the MOST parameterization, they are a natural choice for making part of the model random.

To account for localized bursts of turbulence, the before-mentioned short-tail stability function is extended with multiplicative noise. Using multiplicative noise ensures that the turbulent bursts are temporally localized, allowing the solution to relax towards that of the deterministic model after each burst. Additionally, unlike additive noise, the magnitude of the multiplicative noise depends on the current system state. Moreover, employing multiplicative noise instead of additive noise prevents the stability function from yielding negative values. Lastly, the model for the randomized stability function is chosen such that it includes a time memory to ensure that bursts of turbulence are not dispelled after one time step. The coupled system has the form

$$\mathrm{d}\Delta T = \frac{1}{c_\mathrm{v}}(Q_\mathrm{i} - \lambda \Delta T - \rho c_\mathrm{p} c_\mathrm{D} U \Delta T \phi)\mathrm{d}t \,, \ \Delta T_0 = x_0 \,,$$

$$\mathrm{d}\phi = -r(\phi - f_\mathrm{stab}(R_\mathrm{b}))\mathrm{d}t + \sigma_\phi \phi \, \mathrm{d}W_\phi \,, \ \phi_0 = f_\mathrm{stab}(R_\mathrm{b}(\Delta T_0)) \,, \quad (7)$$

with

$$\sigma_\phi = \begin{cases} 0 & R_\mathrm{b} \leq Ri_\mathrm{c} \\ c_\phi & \text{otherwise} \,, \end{cases}$$

where $W_\phi$ is a Wiener process, $R_\mathrm{b}$ is the bulk Richardson number, $Ri_\mathrm{c} = 0.25$ is the critical Richardson number, and $c_\phi$ is some constant which impacts the noise intensity. The noise is only non-zero for higher Richardson numbers as especially intermittent turbulence in the vSBL regime will be accounted for in the model. The initial condition $\phi_0$ is the value of the short-tail stability function corresponding to $\Delta T_0$. Again, to evaluate how the model reacts to this type of randomization,

in relation to regime transitions, a sensitivity analysis is performed. The setup for the sensitivity analysis is similar to the one in Sect. 2.2.2. That means the model (Eq. 7) is run 500 TS2 times for a combination of $U$ and $c_\phi$ values, and the initial condition $\Delta T_0$ is equal to 4 or TS3 24 K. Then, for each $U$, the minimum value of $c_\phi$ is identified, which ensures that at least 80 % of all 24 h simulations include at least one transition. This minimum value is abbreviated as $c_{\phi,\mathrm{min}}$. TS4

A similar sensitivity study to that in Sect. 2.2.2 and 2.2.3 was performed with 1000 model runs instead of 500 and for transitions from vSBL to wSBL. TS5 The results of this study are not shown here as, for wind speeds less than $5.3\,\mathrm{m\,s^{-1}}$, even for high noise levels, e.g., $c_\phi = 3$, hardly any of the simulations included a transition. Our hypothesis is that this is due to the fact that the model does not have enough degrees of freedom to nonlinearly enhance the effect of the noise. We expect this to be different in a single-column model.

Lastly, the bifurcation-driven transitions of the deterministic model are compared with the noise-induced transitions of the randomized model in Fig. 10. To achieve this, the randomized model (Eq. 7) is run 500 times with an incorporated time-varying wind forcing. The wind speed is modeled as a deterministic step function which increases (left column) or decreases (right column) by $0.1\,\mathrm{m\,s^{-1}}$ roughly every 30 min, as shown by the green curve in panels (a) and (d). The effect of the randomization is studied for both transition types. Hence, the simulations either start in the vSBL state (left column) or the wSBL state (right column). Panels (a) and (d) show the time evolution of the forcing parameter $U$ in green. The black line is the evolution of the bulk Richardson number for one simulation. The gray-shaded area is the region where $R_\mathrm{b} > Ri_\mathrm{c}$. This is the region where noise is added to the stability function. Panels (b) and (e) display the time evolution of $\Delta T$, with the forcing given in panel (a) or (d). The gray lines correspond to 500 realizations of $\Delta T$. The black line is one realization of $\Delta T$ with $R_\mathrm{b}$, as in panel (a) or (d). The mean of all 500 simulations is given by the blue line. The orange line is the solution of the deterministic model (see Eq. 2) with $U$ as given in panel (a) or (d). Panels (c) and (f) have the same color coding as panels (b) and (e), only for $\phi$ instead of $\Delta T$.

On average, the deterministic model and the random simulations have the same transition time for transitions from vSBL to wSBL. However, with a randomized stability function, the transition time is no longer dependent on a specific wind velocity but allows for transitions to occur over a range of velocities before and after the transition of the deterministic model occurs. Therefore, the transition time becomes a range of roughly 4 h instead of a fixed time point. The transitions start within the wind speed range of 5.5 to $6.25\,\mathrm{m\,s^{-1}}$. This observation aligns with the findings presented by (Baas et al., 2019), who demonstrated, through their analysis of observational data from Dome C, that wind speeds below approximately $4\,\mathrm{m\,s^{-1}}$ are associated with highly pronounced inversions ranging from 20 to 25 K. Conversely, wind speeds

exceeding approximately $7\,\mathrm{m\,s^{-1}}$ correspond to comparatively weaker inversions on the order of 5 K. In contrast to vSBL-to-wSBL transitions, transitions from wSBL to vSBL are delayed compared to the deterministic model and occur over a narrow period. Combining the results, it follows that the probability of transitions significantly increases with the use of a randomized short-tail stability function. Therefore, we suggest a modeling compromise where, instead of the conventionally used long-tail stability function, a stochastic parameterization, which includes random bursty features, is used. Further research should assess if this alternative methodology has the potential to enhance the accuracy of large-scale statistics.

## 3 Summary and conclusions

This study expands upon prior research on SBL regime transitions by providing an explanation for the constraints encountered when depicting regime transitions in NWPs. This is achieved by examining the significance of transient phenomena as triggers for abrupt transitions. We used a randomized version of the conceptual model defined by van de Wiel et al. (2017) as an exemplary model to study the sensitivity of the polar SBL to small-scale perturbations and to investigate how related model uncertainty can impact the mean state of the boundary layer. The conceptual model by van de Wiel et al. (2017) is capable of accommodating scenarios with multiple stable equilibria. Therefore, in relation to our objectives, it provides an ideal model for which the theoretical dynamical stability is well understood.

In the first part, we studied the impact of the stability function used in the model on the likelihood of regime transitions in the context of noise-induced tipping. We showed that, for a short-tail stability function, in comparison to a long-tail one, the bistable region is significantly wider. In addition, the potential barrier for the long-tail stability function is shallower, which decreases the chance for abrupt transitions. Combining both results, we concluded that the stability function highly impacts the likelihood of transitions and that, with a short-tail one, the bistability of the system and abrupt transitions are better represented in the model. If NWPs exhibit the regime bistability, the usage of a long-tail stability function would lead to a smaller range of wind speeds for which transitions can occur due to the narrower bistable region. Moreover, the transitions would be smoothed out as a consequence of the shallower potential barrier. In contrast, a randomized short-tail stability function allows for noise-induced abrupt tipping.

In the second part, we analyzed how model uncertainty can be addressed in the conceptual model. We focused on model uncertainty related to unresolved processes, wind forcing, and turbulence parameterization. Firstly, to include small-scale perturbations of an unresolved process in the model, it is extended with additive noise. To assess the model's re-

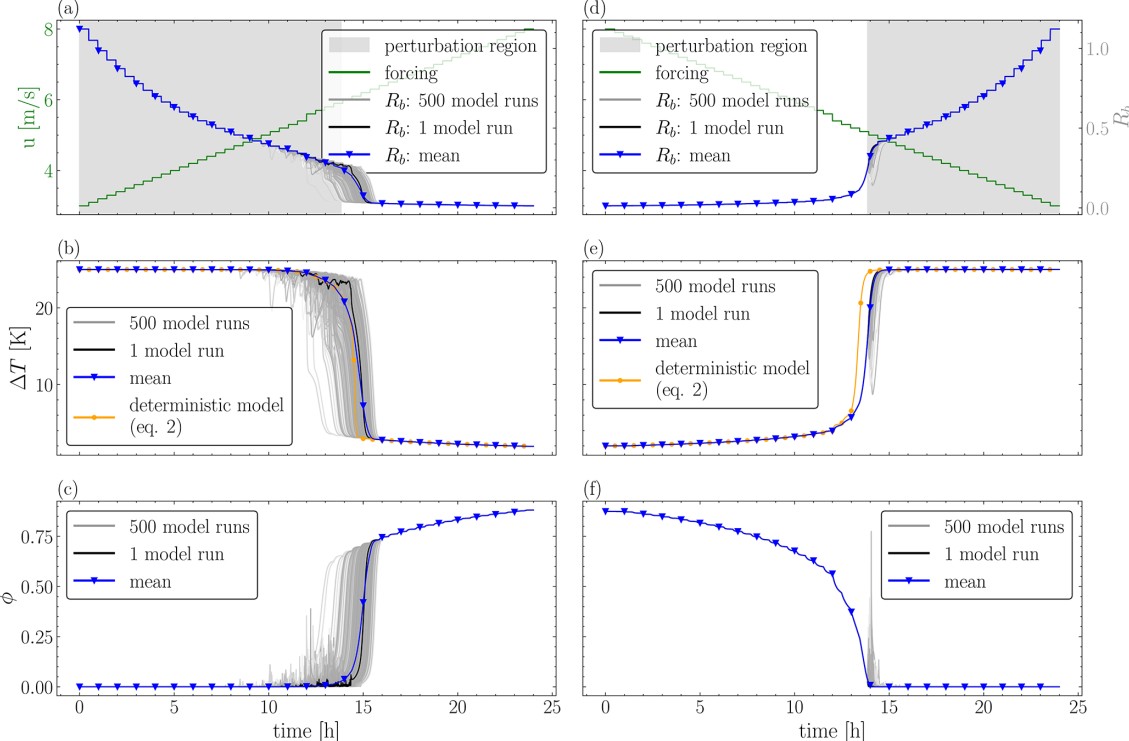

**Figure 10.** Solution of the model with perturbed stability function and variable forcing parameter $U$ (wind velocity). In the left column, the simulations were started in the vSBL state, and $U$ increased. In the right column, the simulations were started in the wSBL state, and $U$ decreased. Panels **(a)** and **(d)** show the time evolution of the forcing, panels **(b)** and **(e)** show the time evolution of $\Delta T$, and panels **(c)** and **(f)** show the time evolution of the stability function.

sponse to this randomization, in terms of regime transitions, a sensitivity analysis was conducted. A Monte Carlo simulation with 500 runs and a combination of different wind speeds and noise strengths was performed. Transitions from wSBL to vSBL and vice versa were studied separately. The sensitivity analysis showed that no noise is required for the system to transition to its equilibrium state (vSBL for low wind speeds and wSBL for high wind speeds). However, to depart from the equilibrium state, higher noise amplitudes are necessary. Within the bistable region, the introduction of noise with small amplitudes allows transitions in both directions. From this, we deduced that the model is highly sensitive to small-scale fluctuations of unresolved processes. Secondly, we studied the effect of including forcing uncertainty in the model by modeling the wind with an Ornstein–Uhlenbeck process. We showed that including randomized wind velocities which seldom exceed the bistable range was not sufficient to induce regime transitions. Therefore, we analyzed the impacts of including randomizations for unresolved processes and wind forcing together. Again, a sensitivity analysis was performed. For low noise amplitudes of the randomized wind velocity, the results were nearly identical to the ones of the model where solely unresolved processes were considered. Lastly, to particularly address model uncertainty related to MOST, we investigated how a

commonly used stability function can be modified to represent unsteady turbulence often present in the SBL. MOST is based on the assumptions of surface homogeneity and turbulence stationarity, which have both been shown to not always be valid. To relax this assumption and allow the representation of unsteady turbulence, we modified the short-tail stability function by van de Wiel et al. (2017) with a time memory and multiplicative noise. This is one way to represent localized turbulence bursts through a stochastic model. Moving forward, a natural progression would be designing a more sophisticated framework where the noise level is linked to the stratification level since unsteady turbulence is known to mainly occur in high stratification. Such an approach has, for example, been pursued by Boyko and Vercauteren (2023b). The authors proposed a stochastic extension to MOST which scales with the Richardson number and implemented it in a single-column model (Boyko and Vercauteren, 2023a). The parameters of the stability correction were estimated based on observational data. A similar approach could be done for NWPs or Earth system models to include localized turbulent bursts. The inclusion of localized turbulence bursts is particularly significant as localized events can trigger regime transitions in the stable boundary layer. A randomized stability function offers an alternative way to mitigate excessive mixing resulting from a long-tail stability function (Baas et al.,

2017) while preventing the system from being trapped in a highly stable state when using a short-tail stability function (Kähnert, 2022). Moreover, the usage of a randomized stability function increases the variability of turbulent mixing, potentially similarly to what an increase in resolution in a NWP would do by resolving more small-scale heterogeneity. Therefore, we hypothesize that a lower resolution could be used for NWP runs but possibly with a higher number of ensemble members. This hypothesis is, for example, supported by the research of Davini et al. (2017) on evaluating the impact of stochastic physics parameterizations. The authors used multiplicative noise to represent model uncertainty due to the parameterization process in the EC-Earth global climate model. For their study, they ran a maximum of 10 ensemble members. The authors demonstrated that the inclusion of stochasticity in the physics parameterizations can be as effective as resolution, and in some cases, it can be even more effective. The need for higher ensemble numbers when the stability function is randomized may be circumvented by its time stochasticity. We showed that, by using a randomized stability function, transitions from wSBL to vSBL are delayed. In contrast, transitions from vSBL to wSBL occurred both before and after the one from the purely deterministic model, thereby increasing the transition period to 4 h. Sun et al. (2012) separate the vSBL and wSBL regimes with a sharp wind speed threshold. With our finding, this sharp threshold is extended to a range of wind speeds for which transitions are possible. Future research will entail studying the inclusion of the same and additional model uncertainties in a higher-order model like a single-column model.

**Code and data availability.** The 30 min averaged Dome C data which are used in this study can be accessed on PANGEA (see https://doi.org/10.1594/PANGAEA.932512, Genthon et al., 2021). The source code to reproduce all results can be found here: https://github.com/BoundaryLayerVercauteren/energy_balance_model (last access: TS6 ; DOI: https://doi.org/10.5281/zenodo.10469086, Kaiser, 2024TS7).

**Author contributions.** All the authors conceptualized the research project. The funding was acquired by NV. The methodology was developed by all the authors. AK primarily performed the analysis and validation with support from NV and SK. AK developed the software. AK prepared the paper. All the authors reviewed and edited the paper.

**Competing interests.** The contact author has declared that none of the authors has any competing interests.

**Disclaimer.** Publisher's note: Copernicus Publications remains neutral with regard to jurisdictional claims made in the text, published maps, institutional affiliations, or any other geographical representation in this paper. While Copernicus Publications makes every effort to include appropriate place names, the final responsibility lies with the authors.

**Acknowledgements.** The authors thank Christophe Genthon and Etienne Vignon for providing the 10 min averaged Dome C data, obtained as part of the CALVA observation project supported by the French polar institute IPEV. Moreover, we wish to thank Peter Ashwin and Terje Koren Berntsen for the helpful discussions. In addition, we thank the anonymous reviewer and Adam Monahan for their valuable and constructive feedback, which helped with improving our paper.

**Financial support.** This project has received funding from the European Union's Horizon 2020 research and innovation program under the Marie Skłodowska-Curie grant agreement no. 945371.

**Review statement.** This paper was edited by Wansuo Duan and reviewed by Adam Monahan and one anonymous referee.

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

## Remarks from the typesetter

TS1    Please note: subscript format changed to roman. According to our standards, all subscripts that have two or more letters, or are abbreviations for a word, have to be roman.

TS2    Please give an explanation of why this needs to be changed. We have to ask the handling editor for approval. Thanks.

TS3    Please give an explanation of why this needs to be changed. We have to ask the handling editor for approval. Thanks.

TS4    Please give an explanation of why this needs to be changed. We have to ask the handling editor for approval. Thanks.

TS5    Please give an explanation of why this needs to be changed. We have to ask the handling editor for approval. Thanks.

TS6    Please provide date of last access.

TS7    Please confirm added citation.

TS8    Please confirm reference list entry.

TS9    Please confirm addition.

TS10   Please confirm.