# Peer review of "Sensitivity of the polar boundary layer to transient phenomena"

_EGUsphere, 2023_

## Author Comment (AC1)

**NPG-2023-1519: Reply to comments from Referee #1**

Below, you'll find our detailed responses addressing the reviewer's feedback. We truly appreciate the valuable input provided on our work. The initial comments are presented in bold font, followed by our responses in standard font.

**1. The concept of potential was introduced at the end of section 2.2 and is only utilized in figure 2. However, this concept is not employed in the remainder of the paper. Could you please provide a more thorough explanation of the physical interpretation of this concept and how it relates to the other results?**

We agree that a more detailed explanation of the potential is beneficial and will add it to the revised version.

**2. The legends for figures 1 and 10 are very small and incomplete.**

Thank you for pointing this out. We will fix that when we submit the revised version.

**3. As a suggestion to improve the readability of the paper, I propose that section 2.3 be integrated into section 2.2 ("Impact of the choice of the stability function"). This consolidation would be beneficial since the results presented in section 2.2 involve randomized wind speeds. Alternatively, moving section 2.2 to section 2.3 would eliminate the need for readers to navigate back and forth in the manuscript.**

We agree that the structure of the section 2 can be improved and will move section 2.2 to 2.3, as you suggested, to improve the readability. In addition, we will briefly introduce the randomizations (e.g. equation 3) in section 2.3 and give the current sections 2.3.1-2.3.3 more descriptive names.

We realize that we made a mistake in the description of figure 3. The additive noise model (equation 3, internal variability) has been used not the randomized wind speed model to create the histograms. We sincerely apologize and will correct the description of figure 3 in the revised version.

**4. One curiosity: What is the computational cost of the method to randomize the stability function for application in NWP? Is it necessary to perform an ensemble of simulations or just a single simulation for the climate and weather models?**

In principle, an ensemble of simulations is needed to see the multi-modality, as mentioned in the paper, in the output data and to get accurate regime occupation statistics. The answer to how many ensemble runs are required to achieve this depends on the goal and requires further research.

We expect large benefits from the inclusion of stochasticity, more in terms of sampling the variability, rather than improving the representation of the mean state. The usage of a randomized stability function potentially increases the variability of turbulent mixing, similar to what an increase in resolution would do by resolving more small-scale heterogeneity. Therefore, we hypothesize that a lower resolution may be required, but possibly a higher ensemble number would be needed. This hypothesis is, for example, supported by the research of Davini et.al. (2017) on evaluating the impact of stochastic physics parameterizations. They used multiplicative noise to represent model uncertainty due to the parameterization process in the EC-Earth global climate model. For their study, they ran a maximum of 10 ensemble members. The authors showed that the inclusion of stochasticity in the physics parameterizations can be as effective as resolution, and in some cases even more effective. In addition, they mention that with their setup the 16km resolution model can be run 9-10

times with the same amount of core hours as the 30km resolution one. Furthermore, the stochastic stability function we use is randomized in time. This might reduce the number of required ensemble members, as the time variability may act to sample appropriate statistics of mixing dynamics. Generally, the computational cost of a stochastic approach with higher ensemble number but lower resolution is potentially less than that of a high-resolution simulation, as the stochastic computations can be run independently and hence are highly parallelizable.

References:

Davini, P., and Coauthors, 2017: Climate SPHINX: evaluating the impact of resolution and stochastic physics parameterisations in the EC-Earth global climate model. Geoscientific Model Development, 10, 1383–1402, https://doi.org/10.5194/gmd-10-1383-2017.

**5. line 120 - ODE is not defined.**

Thank you for your comment. We will replace the abbreviation, ODE, with ordinary differential equation.

**6. line 120 - "ΔT, between a reference height (Tr) and the surface temperature (Ts)" -> add "Temperature inversion".**

We will add this to the revised version.

**7. line 125 - ΔT is defined in line 125, and is no longer necessary.**

Thanks for pointing this out. We will remove the duplicated definition of $\Delta T$.

**8. line 145, table 1 - units should not be in italics.**

This will be fixed in the revised version.

---

## Author Response (AR1)

**Changes to EGUSPHERE-2023-1519**

Dear editor, dear reviewers,

We would like to thank the editor, the anonymous reviewer and Adam Monahan for the time and efforts devoted to our manuscript. We are grateful for the constructive feedback on our work. Please find below our point-by-point answer to the editors and reviewers' comments. The original comments are in *italic font*, followed by our answers in regular font. Changed text is marked in blue.
* * *
Editor:

1. *For the next revision, I kindly ask you to remove the text part "Copyright statement. TEXT" on page*

   Thank you for pointing this out. We have removed this text.

2. *Regarding your figures, please keep colour blindness in mind and avoid the parallel usage of green and red (see F7). For a list of colour scales that are illegible to a significant number of readers, please visit ColorBrewer 2.0 (*http://www.colorbrewer2.org/*).*

   Thank you for drawing our attention to this. We have updated the colors in figure 7 accordingly.
* * *
Reviewer #1 (Anonymous):

*The manuscript analyzes transitions in the stable boundary layer (SBL) regime from weakly stable to very stable boundary layers, and vice versa. The analysis is based on a conceptual model proposed by van de Wiel et al. (2017) with inclusion of a stochastic model to account for the effects of small-scale fluctuations of unresolved processes in the models. The study holds the potential to offer novel insights. However, I encountered some challenges while reading the manuscript due to its organization. The manuscript would require minor revisions to meet the standards for publication in EGUsphere.*

We thank the reviewer for the review of our work and the positive assessment. The comments are answered below.

**Major comments:**

1. *The concept of potential was introduced at the end of section 2.2 and is only utilized in figure 2. However, this concept is not employed in the remainder of the paper. Could you please provide a more thorough explanation of the physical interpretation of this concept and how it relates to the other results?*

   We agree that a more detailed explanation of the potential is beneficial. Therefore, we have added the following text to the introduction:

"This stochastic model can then be reinterpreted as a gradient flow, in which the temperature inversion evolves according to an underlying energy potential and the equilibrium points correspond to minima of this potential. If several minima exist, then the different equilibria are separated by a local maximum that can be interpreted as a potential barrier that needs to be crossed if the system is to transition between the different equilibria. We highlight that the choice of the stability function has a large impact on the height of the potential barrier and hence on the energy input needed to undergo a regime transition."

Moreover, we have changed the text in the section "Impact of the choice of the stability function" to:

"Secondly, to explain why the likelihood of transition is very dependent on the choice of stability function we introduce the concept of a potential. For this equation (2) can be rewritten as a gradient system $\frac{\Delta T}{dt} = -V'(\Delta T), \quad \Delta T(t_0) = \Delta T_0$ where $V: \mathbb{R} \to \mathbb{R}$ is an underlying potential influencing the dynamics of the temperature inversion. The extrema of the potential V correspond to the equilibria of $\Delta T$, i.e. for an equilibrium point $\Delta T_e$ it holds $V'(\Delta T_e) = 0$. In general, the dynamics of $\Delta T$ will evolve towards the nearest local minimum of the potential. If it resides there, signifying a stable equilibrium, it would require the addition of significant random fluctuations to exit this state. Indeed, if a local maximum separates two local minima, i.e. two possible stable equilibria, the difference between the potential's minimum and maximum is an energy depth that the dynamics has to overcome in order to transition to a second stable equilibrium. This is called a potential barrier. In our context, that means that if the wind forcing is such that both vSBL and wSBL are supported solutions, the potential barrier describes the intensity of fluctuations of $\Delta T$ that are needed to transition between the two states, assuming no other changes in the forcing or dynamics."

2. *The legends for figures 1 and 10 are very small and incomplete.*

Thank you for pointing this out. We have fixed that in the revised version.

3. *As a suggestion to improve the readability of the paper, I propose that section 2.3 be integrated into section 2.2 ("Impact of the choice of the stability function"). This consolidation would be beneficial since the results presented in section 2.2 involve randomized wind speeds. Alternatively, moving section 2.2 to section 2.3 would eliminate the need for readers to navigate back and forth in the manuscript.*

We agree that the suggested changes would enhance the clarity of the text and that the structure of section 2 can be improved. Therefore, we have moved section 2.2 to 2.3, as suggested, to improve the readability. In addition, we briefly introduced the randomizations (e.g. equation 3) in (the original) section 2.3 and gave the current sections 2.2.2.-2.2.4. more descriptive names.

We realize that we made a mistake in the description of figure 3. The additive noise model (equation 3, internal variability) has been used not the randomized wind speed model to create the histograms. We sincerely apologize and have corrected the description of figure 3 in the revised version.

4. *One curiosity: What is the computational cost of the method to randomize the stability function for application in NWP? Is it necessary to perform an ensemble of simulations or just a single simulation for the climate and weather models?*

Thank you for this interesting question. In principle, an ensemble of simulations is needed to see the multi-modality, as mentioned in the paper, in the output data and to get accurate regime occupation statistics. The answer to how many ensemble runs are required to achieve this depends on the goal and requires further research.

We expect large benefits from the inclusion of stochasticity, more in terms of sampling the variability, rather than improving the representation of the mean state. The usage of a randomized stability function potentially increases the variability of turbulent mixing, similar to what an increase in resolution would do by resolving more small-scale heterogeneity. Therefore, we hypothesize that a lower resolution may be required, but possibly a higher ensemble number would be needed. This hypothesis is, for example, supported by the research of Davini et.al. (2017) on evaluating the impact of stochastic physics parameterizations. They used multiplicative noise to represent model uncertainty due to the parameterization process in the EC-Earth global climate model. For their study, they ran a maximum of 10 ensemble members. The authors showed that the inclusion of stochasticity in the physics parameterizations can be as effective as resolution, and in some cases even more effective. In addition, they mention that with their setup the 16km resolution model can be run 9-10 times with the same amount of core hours as the 30km resolution one. Furthermore, the stochastic stability function we use is randomized in time. This might reduce the number of required ensemble members, as the time variability may act to sample appropriate statistics of mixing dynamics. Generally, the computational cost of a stochastic approach with higher ensemble number but lower resolution is potentially less than that of a high-resolution simulation, as the stochastic computations can be run independently and hence are highly parallelizable.

As we found your comment very interesting, we have added to following text to the discussion section:

"Moreover, the usage of a randomized stability function potentially increases the variability of turbulent mixing, similar to what an increase in resolution in a NWP would do by resolving more small-scale heterogeneity. Therefore, we hypothesize that a lower resolution may be required for NWP runs, but possibly a higher ensemble number would be needed. This hypothesis is, for example, supported by the research of Davini et.al. (2017) on evaluating the impact of stochastic physics parameterizations. The authors used multiplicative noise to represent model uncertainty due to the parameterization process in the EC-Earth global climate model. For their study, they ran a maximum of 10 ensemble members. The authors demonstrated that the inclusion of stochasticity in the physics parameterizations can be as effective as resolution, and in some cases even more effective."

References:

Davini, P., von Hardenberg, J., Corti, S., Christensen, H. M., Juricke, S., Subramanian, A., Watson, P. A. G., Weisheimer, A., and Palmer, T. N.: Climate SPHINX: evaluating the impact of resolution and stochastic physics parameterisations in the EC-Earth global climate model,

Geoscientific Model Development, 10, 1383–1402, https://doi.org/10.5194/gmd-10-1383-2017, publisher: Copernicus GmbH, 2017.

**Minor comments:**

1. *line 120 - ODE is not defined.*

   We have replaced the abbreviation, ODE, with ordinary differential equation.

2. *line 120 - "ΔT, between a reference height (Tr) and the surface temperature (Ts)" -> add "Temperature inversion".*

   We have added this to the revised version.

3. *line 125 - ΔT is defined in line 125, and is no longer necessary.*

   Thanks for pointing this out. We have removed the duplicated definition of $\Delta T$.

4. *line 145, table 1 - units should not be in italics.*

   This has been fixed.
* * *
Reviewer #2 (Adam Monahan):

*This study considers fluctuation-induced transitions between very (vSBL) and weakly (wSBL) stable regimes in the stably stratified atmospheric boundary layer (SBL) in the context of a well-established idealized low-dimensional energy budget model. Three different classes of variability are modelled by stochastic processes: general unresolved physical processes as an additive noise, fluctuating wind speed as a multiplicative Ornstein-Uhlenbeck process (OUp), and deviations from Monin-Obukhov similarity theory (MOST) due to non-equilibrium turbulence in the VSBL as another multiplicative OUp. It is found that combining external fluctuations with a short-tailed stability function produces more abrupt regime transitions than when a long-tailed stability function is considered.*

*My recommendation is that the study is acceptable for publication in Nonlinear Processes in Geophysics after major revisions.*

Thank you for the constructive and helpful feedback on our work. We answer the specific points below.

**Major comments:**

1. *Much of the analysis in the study contrasts the use of short-tailed and long-tailed stability functions. An essential question in the choice of stability function to model turbulent transport is the relevant horizontal scale represented by the model. Long-tailed stability functions were introduced in the context of numerical weather prediction/Earth system modelling to account for gridboxes that are much larger than the horizontal scale of wSBL/vSBL patches, so fluxes computed over an individual grid box include contributions from regions of both states. While use of Dome C station data to guide model development indicates that the model is to be interpreted locally, motivation for the analysis is presented in terms of NWP/ESM biases. The revised study should*

*carefully frame the analysis in terms of the relevant horizontal scales – and discuss the importance of this choice.*

It is true that the use of long-tail stability functions is often justified by a lack of resolution, as stated in the review paper by Sandu et al (2013): "It is often argued that the artificial enhancement of the mixing in stable conditions is needed to account for contributions to vertical mixing associated with surface heterogeneity, gravity waves, or mesoscale variability that are not explicitly represented in models." In that sense, with higher horizontal resolution, one could expect less need to use long-tail stability functions as more processes leading to vertical mixing are resolved. However, we believe that the interpretation that this is introduced to represent gridboxes with contributions from weakly stable (wSBL) and very stable (vSBL) patches is slightly misleading. Indeed, the wSBL and vSBL states are regimes of the full boundary layer. Within the vSBL regime, the existence of temporally and spatially intermittent turbulence and a plethora of submeso scale motions is known, and these all lead to localized perturbations of the dynamics that prevent a complete collapse of turbulence. Long-tail stability functions were mainly introduced to prevent this collapse, and also to tune larger-scale model scores.

Our hypothesis is that the usage of a randomized stability function can represent the spatial and temporal variability. This potentially increases the variability of turbulent mixing, similarly to what an increase in resolution in a NWP would do by resolving more small-scale heterogeneity. Therefore, we hypothesize that a lower resolution may be required for NWP runs, but possibly a higher ensemble number would be needed. This hypothesis is, for example, supported by the research of Davini et.al. (2017) on evaluating the impact of stochastic physics parameterizations. The authors used multiplicative noise to represent model uncertainty due to the parameterization process in the EC-Earth global climate model. For their study, they ran a maximum of 10 ensemble members. The authors demonstrated that the inclusion of stochasticity in the physics parameterizations can be as effective as resolution, and in some cases even more effective.

To address the importance of the horizontal resolution we have added the following text to the discussion section:

"Moreover, the usage of a randomized stability function potentially increases the variability of turbulent mixing, similar to what an increase in resolution in a NWP would do by resolving more small-scale heterogeneity. Therefore, we hypothesize that a lower resolution may be required for NWP runs, but possibly with a higher number of ensemble members. This hypothesis is, for example, supported by the research of Davini et.al. (2017) on evaluating the impact of stochastic physics parameterizations. The authors used multiplicative noise to represent model uncertainty due to the parameterization process in the EC-Earth global climate model. For their study, they ran a maximum of 10 ensemble members. The authors demonstrated that the inclusion of stochasticity in the physics parameterizations can be as effective as resolution, and in some cases even more effective. The need for higher ensembles numbers when the stability function is randomized may be circumvented by its time stochasticity."

Regarding your comment about the usage of the Dome C data to guide model development, our paper describes a sensitivity study of the idealised boundary layer rather than the development of a new model. We now clarified this point in the introduction, to avoid misunderstanding of the scope. For the sensitivity analysis we use model parameters which were estimated based on the Dome C data by van de Wiel et al. (2017). The only other case where we use Dome C data in our model is to approximate $\sigma_U$ to ensure using a relevant order of magnitude of the perturbations.

References:

Davini, P., von Hardenberg, J., Corti, S., Christensen, H. M., Juricke, S., Subramanian, A., Watson, P. A. G., Weisheimer, A., and Palmer, T. N.: Climate SPHINX: evaluating the impact of resolution and stochastic physics parameterisations in the EC-Earth global climate model, Geoscientific Model Development, 10, 1383–1402, https://doi.org/10.5194/gmd-10-1383-2017, publisher: Copernicus GmbH, 2017.

Sandu, I., Beljaars, A. C. M., Bechtold, P., Mauritsen, T., and Balsamo, G.: Why is it so difficult to represent stably stratified conditions in numerical weather prediction (NWP) models?, Journal of Advances in Modeling Earth Systems, 5, 117–133, https://doi.org/10.1002/jame.20013, 2013.

van de Wiel, B. J. H., Vignon, E., Baas, P., van Hooijdonk, I. G. S., van der Linden, S. J. A., Antoon van Hooft, J., Bosveld, F. C., de Roode, S. R., Moene, A. F., and Genthon, C.: Regime Transitions in Near-Surface Temperature Inversions: A Conceptual Model, Journal of Atmospheric Sciences, 74, 1057–1073, https://doi.org/10.1175/JAS-D-16-0180.1, 2017.

2. *The results of van de Wiel et al. (2017) and Ramsey and Monahan (2022) indicate that conditions in which the PBL is bistable are relatively rare outside the very high latitudes. However, the title of the manuscript refers to "nocturnal and polar" boundary layers. The revised manuscript should explain the emphasis on bistable conditions to the exclusion of more common conditions with single (deterministic) fixed points in which stochastic fluctuations can still induce rapid transitions. I also recommend the title be revised to better reflect the parameter space on which the study focuses.*

We agree that the original title was inaccurate, and we have changed it to "Sensitivity of the polar boundary layer to transient phenomena". In addition, we have emphasized in the introduction and model description section that we focus on the Polar SBL. We have for example added the following text to the introduction:

"The focus hereby is on the Polar night and therefore the model parameter values are chosen such that they represent a Polar context."

and

"The same values as in van de Wiel et al. (2017) (Table 1, Dome C) are used for all parameters unless stated otherwise. The focus is on parameters representing a Polar context as studies have shown that in these regions bistability exists (Ramsey and Monahan (2022), van de Wiel et al. (2017)) and hence transitions can be more abrupt. Therefore, in this context the impact of small-scale perturbations for regime transitions is especially relevant."

to the model description section.

3. *Following on from the previous point, I note that for systems driven by multiplicative OUp noise extrema of stationary pdfs can exist that do not correspond to deterministic fixed points (cf. Monahan 2002). This effect can be understood in terms of a dynamical system driven by Brownian motion in the expanded state space corresponding to the original state variable and the OUp. The authors may find this perspective useful for interpreting their results.*

This is an interesting comment. The following text has been added to the section 2.2.3.:

"It shall be noted that the model with multiplicative noise may have different equilibrium points than the deterministic model (see chapter 5.4 Pavliotis (2014)). This for example has been seen in Bashkirtseva et al. (2015) and Monahan (2002). Nonetheless, comparable with the results of model (3), where solely unresolved processes were considered while disregarding fluctuating wind speed, figure 8 shows that no noise is required for the system to transition to the equilibrium state of the deterministic model (vSBL for low wind speeds and wSBL for high wind speeds)."

References:

Bashkirtseva, I., Ryazanova, T., and Ryashko, L.: Stochastic bifurcations caused by multiplicative noise in systems with hard excitement of auto-oscillations, Physical Review E, 92, 042 908, https://doi.org/10.1103/PhysRevE.92.042908, publisher: American Physical Society, 2015.

Monahan, A. H.: Stabilization of Climate Regimes by Noise in a Simple Model of the Thermohaline Circulation, Journal of Physical Oceanography, 32, 2072–2085, https://doi.org/10.1175/1520-0485(2002)032<2072:SOCRBN>2.0.CO;2, publisher: American Meteoro-510 logical Society Section: Journal of Physical Oceanography, 2002.

Pavliotis, G. A.: Stochastic Processes and Applications, vol. 60, Springer New York, https://doi.org/10.1007/978-1-4939-1323-7, 2014.

4. *A novel aspect of the study is the focus on probability distributions from 24-hour simulations from fixed initial conditions, rather than the long-term statistical equilibrium stationary distributions. What motivates the choice of the 24-hour period? If the focus is on the polar night, the effect of the diurnal cycle will be weak (other than through the Coriolis force acting on the flow above the shallow SBL, which is neglected in the model under consideration). The revised study should provide a justification for this aspect of the experimental design.*

This choice was poorly explained indeed. Essentially, we wanted to have long enough simulations to reach a quasi-equilibrium state but replaced the temporal statistical equilibrium with a Monte-Carlo sampling study. The idealised model is forced by a constant wind speed, and this one would never stay constant for more than a few hours in reality, as it will vary with synoptic conditions. This is why we decided to have shorter simulations, but a Monte-Carlo approach to have good statistics. The choice of 24 hours was partly for practical reasons as we focus on a grid search combined with Monte Carlo simulations which requires a significant amount of computing power. Our focus was on having enough Monte Carlo simulations such that the statistics did not change significantly by a further increase in simulations.

The following text has been added to the section randomization strategies:

"The simulation time for every model run should be long enough to reach a quasi-equilibrium state. To achieve this rather than simulating until a temporal statistical equilibrium is reached the Monte-Carlo sampling study is performed. As the idealized model is forced by a constant wind speed which would vary with synoptic conditions and not be constant for more than a few hours we deemed a simulation time of 24 hours with time steps of 1 second length a decent

compromise. In addition, this choice was also for practical reasons as the focus in the following sections is on a grid search combined with Monte Carlo simulations which requires a significant amount of computing power. The Monte Carlo simulations are run with 500 realizations. A comparison with 1000 simulations showed similar results. Hence, 500 realizations are deemed sufficient for the rest of the sensitivity analysis."

5. *Throughout the text statements are made along the lines of "outside of the bistable range low noise values are required for the system to approach the single equilibrium state". By definition, and as is evident from the corresponding Figures, this noise level is zero. The associated statements in the revised manuscript should be consistent with this fact.*

We have changed statements like this to "no noise is required to transition" or similar formulations.

6. *LL 258-261: These results are not surprising and could have been predicted without the need of any numerical simulations. The revised manuscript should replace this text with statements regarding what new has been learned from the simulations, or clearly state that the cited results are expected from first principles.*

We agree that the results are not surprising but for completeness we want to keep them. To reflect that we changed the formulation from "based on this discussion" to "as expected".

7. *Section 2.3.2: I am confused regarding the estimation of $\sigma_U$. The text following Eqn. (4) indicates that variations in U are intended to represent submesoscale processes. Are the Dome C data filtered to exclude processes on mesoscale and longer timescales? If these other processes are included in the estimate of $\sigma_U$ it is perhaps not surprising that the variance is larger than had been expected. The revised manuscript should include a clearer description of the estimation of $\sigma_U$.*

Thank you for your comment. Excluding mesoscale and longer timescales from the data for the estimation of $\sigma_U$ was very helpful and we had overlooked this. By using moving average filtering, as you suggested, with a window length of 60 minutes we get a new estimate for $\sigma_U$ for the Dome C data which is 0.03. This is significantly smaller than our estimate which was calculated without timescale filtering. If $\sigma_U = 0.03$ and $\sigma_i = 0.0$ 97% of the 500 simulations include a transition and 34% of the wind velocity values are on average outside the bifurcation region. These values were calculated for simulations starting the vSBL state. Therefore, we have rerun the sensitivity analysis with $\sigma_U = 0.01$ and 0.03. We have chosen to include $\sigma_U = 0.01$ as this is the smallest value for which hardly any of the wind velocity values are outside of the bifurcation region, i.e. on average less than 1%. But for $\sigma_U = 0.01$ none of the 500 simulations include a transition.

To reflect these changes, we have adapted the text as follows:

"The value for $\sigma_U$ is chosen based on 30-minute averaged observational data from Dome C (Genthon et al., 2021). In their study Baas et al. (2019) defined the bistable region for Dome C as 4 ms−1 ≤ U ≤ 7 ms−1. The same thresholds and data from the year 2013, which has a mean value for the wind speed of 5.6 ms−1, is used get an estimate of $\sigma_U$. To exclude mesoscale and longer timescales the data is filtered with moving average filtering with a window length of 60 minutes. For this dataset $\sigma_U$ is equal to 0.03 ms−3/2. Choosing $\sigma_U$ this big leads to high variations in U and not to the small-scale perturbations we are interested in. In fact, while 97% of all simulations of a

Monte Carlo run with 500 simulations include a transition 34% of the wind velocity values are on average outside the bifurcation region which contradicts the assumption of small-scale perturbations. One example of a run with $\sigma_U$ = 0.03ms−3/2 is shown in figure 7 b). Additionally, $\sigma_U$ = 0.01 ms−3/2 is considered as this is the smallest value for which hardly any of the wind velocity values are outside of the bifurcation region, i.e. on average less than 1%. But for this $\sigma_U$ none of the 500 simulations include a transition (see figure 7 a)."

8. *The legend of Figure 10 is clearly wrong (or is rendering incorrectly on my version of Adobe Professional), and the caption only describes the left column. The Figure should be corrected and the caption updated in the revised manuscript.*

The legend is correct but seems to be only partially rendered in the web browser. For us the figure is displayed correctly when we download the pdf.

We agree that the caption of figure 10 and the corresponding text can be improved by better explaining the difference between the left and right column.

The new caption is:

"Solution of the model with perturbed stability function and variable forcing parameter U (wind velocity). In the left column the simulations were started in the vSBL state and U increases. In the right column the simulations were started in the wSBL state and U decreases. Panel a) and d) show the time evolution of the forcing, panel b) and e) of ΔT, and c) and f) of the stability function."

In addition, we have added a legend to every subplot of figure 10.

9. *Section 2.3.3: The first set of results in this section show that on their own, multiplicative fluctuations of the stability function do not induce transitions between states. This is an interesting result that deserves further discussion. Later in the section, when both wind speed and similarity function fluctuations are considered, transitions are attributed to variations of the stability function (e.g. L 362, "However, with a randomized stability function …"). I do not follow this conclusion – in the presence of two sources of variability how do the authors attribute transitions to only one? Finally, I do not understand how the authors conclude that " … it follows that the probability of transitions significantly increases with the use of a randomized short-tailed stability function." This statement seems to directly contradict what is said earlier in the section. These issues should be addressed in the revised manuscript.*

Thank you for your comment but this is likely a misunderstanding. The wind is not randomized in this case. It is rather a deterministic function where u is a step function with increasing values if the simulations are started in the vSBL state and decreasing values if they are started in the wSBL state. To better reflect this and to prevent misunderstandings the text has been changed to:

"To achieve this the randomized model (6) is run 500 times with an incorporated time-varying wind forcing. The wind speed is modelled as a deterministic step function which increases (left column) or decreases (right column) by 0.1 m/s roughly every 30 minutes as shown by the green curve in panel a) and b)." (section 2.2.4.)

**Minor comments:**

1.  *Table 1: There is a question mark in the units for $c_v$ that should be removed.*

    Thanks for pointing this out. This has been fixed.

2.  *LL 194-195: The text should note here that the distributions come from 24h simulations.*

    We have added this to the text

3.  *LL 201-203: Why "under-representation"? Certainly the number of vSBL states for long-tailed stability functions is smaller than for short-tailed functions but in neither case is there a clear "truth" against which to compare. I recommend revising the text accordingly.*

    We agree that the wording was incorrect and therefore we removed that part of the sentence.

4.  *Figure 3 (and similar): The distributions presented are not densities as they are clearly not normalized. Rather these appear to be bin counts. I recommend replotting as normalized densities or revising the label/caption.*

    The y axes labels in figure 3, 6 and 9 were changed from density to histogram.

5.  *The noise intensities $\sigma$ and decay timescales r both have units. These units should be included in figure axes and when values are quoted in the text.*

    The units have been added in the revised version of the paper.

6.  *The authors may consider combining Figs 6 and 9 to facilitate direct comparison of distributions.*

    We see the point that combining 6 and 9 allows easier comparison but we prefer to keep them separate as they belong to different sections. If they were combined the reader would need to scroll a lot read the description belonging to either the (former) figure 6 or 9.

*I appreciate the importance of idealized model studies such as this one and look forward to seeing a revised manuscript.*

We again thank the editor, the anonymous reviewer and Adam Monahan for their thorough revision. The comments were very valuable and helped with improving our paper.